# Impact of Pharmacist Educational Intervention on Costs of Medication with Improved Clinical Outcomes for Diabetic Patients in Various Tertiary Care Hospitals in Malaysia: A Randomized Controlled Trial

**DOI:** 10.3390/healthcare13080901

**Published:** 2025-04-14

**Authors:** Muhammad Zahid Iqbal, Saad S. Alqahtani, Sara Shahid, Khalid M. Orayj

**Affiliations:** 1Department of Clinical Pharmacy, College of Pharmacy, King Khalid University, Abha 61421, Saudi Arabia; ss.alqahtani@kku.edu.sa (S.S.A.); korayg@kku.edu.sa (K.M.O.); 2Department of Pharmacy Practice, Faculty of Pharmaceutical Sciences, Lahore University of Biological & Applied Sciences, Lahore 53400, Pakistan; sarashahid498@gmail.com

**Keywords:** diabetes mellitus, cost of medication, glycated hemoglobin (HbA1c), pharmacist-led educational intervention, lifestyle modification

## Abstract

Background and Objective: A lifestyle-associated disease, diabetes mellitus, mandates compliance with established policies by physicians and patients to achieve optimal glycemic control. Collaborative care from health care providers and patients is essential for effective management, which slows disease progression, improves quality of life, and reduces medication costs. This study assessed the effectiveness of pharmacist-led educational initiatives provided to patients on clinical outcomes and direct treatment costs for those with diabetes in two public hospitals in Malaysia. Methods: Four hundred type 2 diabetes patients included in this study were randomly allocated to two corresponding groups. The control group, consisting of 200 patients (100 from each hospital), received standard treatment using the Malaysian Clinical Practice Guideline 2015, while the intervention group, also comprising 200 patients (100 from each hospital), received pharmacist-led care through Diabetic Medication Therapy Adherence Clinics (DMTACs), including pharmacist-provided education, alongside conventional treatment. The patients were equally selected from both hospitals based on the sample size calculation. The pharmacists provided educational interventions emphasizing dietary adjustments, lifestyle modifications, the significance of physical activity, and appropriate medication storage. Among these, 143 control patients and 156 intervention patients completed this one-year study, which comprised an initial and two follow-up visits. Clinical outcomes and treatment expenses were evaluated, and a data analysis was performed utilizing version 24 SPSS. Descriptive statistics were presented as the mean ± standard deviation, including normality assessed using the skewness, kurtosis, and Kolmogorov–Smirnov test. Independent *t*-tests were applied for hypothesis testing when the data showed normal distribution. Paired *t*-tests were used for cost assessments. Results: After the research, the group receiving intervention had a much higher decrease in HbA1c levels relative to the control group (3.59% versus 2.17%; *p* < 0.001). The intervention group had considerable decreases in systolic blood pressure (9.29 mmHg) and similarly in diastolic blood pressure (7.58 mmHg; with *p* < 0.005). Additionally, the levels of cholesterol in the intervention group improved significantly (0.13 mmol/L; *p* < 0.001). Moreover, treatment expenses for the pharmacist-led intervention group showed a substantial reduction (*p* < 0.001). By the second follow-up, the additional cost per patient since baseline was MYR 236.07 (Malaysian Ringgit), approximately 53.45 USD, in the control group, compared to only MYR 47.33 per patient, approximately 10.72 USD, in the intervention group with pharmacist involvement. Only medication costs were considered, and all unnecessary medications were discontinued as patient clinical outcomes improved sufficiently with pharmacist intervention, allowing for management through lifestyle changes alone. Counseling costs were not included since the pharmacists providing education were already employed in these hospitals, and no additional pharmacists were appointed for this purpose. Conclusion: Pharmacist-led interventions led to a significant improvement in HbA1c levels. While medication expenses increased in both groups from the initial follow-up, the control group exhibited a significantly greater increase in costs and HbA1c levels than the intervention group.

## 1. Introduction

The impact of pharmacists is extensively documented worldwide in ensuring the suitability and efficacy of pharmacotherapy, as well as enhancing patient adherence to treatment regimens [1]. The lifestyle-related condition type 2 diabetes requires rigorous glycemic regulation. Control relies not only on physician adherence to clinical guidelines for managing disease but also significantly on patient compliance with prescribed pharmaceutical regimens [2]. Patient adherence to prescribed medicine, precise dosing schedules, proper usage, and the appropriate storage of insulin devices are essential considerations for achieving compliance. The compliance of patients can be enhanced through proficient counseling and education delivered by health care experts, as the consequences of unmanaged diabetes lead to numerous issues [2]. Global studies demonstrate that health care practitioners’ engagement in patient education results in improved outcomes in type 2 diabetes mellitus management [3]. However, it remains essential to assess the diverse elements of pharmacist-led interventions through patient counseling on every dimension and outcome of the type 2 diabetes condition.

Currently, there is no known cure for type 2 diabetes mellitus (T2DM). However, the condition can be managed to slow down the progression of diabetic complications and enhance the quality of life for patients [4]. Maintaining proper glycemic control is essential to dropping the risk of microvascular and macrovascular complications in diabetic patients [5]. The primary goal of T2DM management is to minimize their occurrence [6]. The global prevalence of T2DM is steadily increasing [7], with adults in developing countries being at a particularly higher risk of developing the condition [8]. The global prevalence of diabetes continues to rise annually and is projected to reach up to 642 million cases by 2040 [9]. Approximately 90% of all diabetes cases belong to T2DM and are contributing to a growing economic burden on global health care systems [10]. The escalating prevalence of T2DM is placing a growing financial strain on health care systems worldwide, making prevention the ultimate solution, which has proven to be both effective and cost-efficient [11].

Cost-effectiveness in diabetes management is a major concern, as adherence to medication increases the cost of drug treatment but can significantly reduce expenses related to diabetes complications and hospital admissions [12]. Studies indicate that poorly managed diabetes results in increased health care expenditures due to the high costs of treating complications, such as cardiovascular disease, neuropathy, and renal failure [13,14]. Preventing complications through effective glycemic control is a cost-effective strategy [11], but access to affordable treatment options remains a challenge [15]. A 1% reduction in HbA1c levels has been associated with a 21% reduction in risk for microvascular complications [16], underscoring the importance of proactive diabetes management.

Health care expenditure on diabetes treatment is substantial. In 2010, global spending on diabetes and its complications amounted to USD 376 billion, with projections estimating an increase to USD 490 billion by 2030 [17]. In Malaysia, multiple studies have estimated the economic burden of diabetes care. A study by Sharifa Ezat et al. (2009) reported that inpatient diabetes treatment costs MYR 1951.00 per year at Ministry of Health facilities without specialists, increasing to MYR 1974.44 annually when specialists are available [18]. The cost of ambulatory care per patient without complications is approximately MYR 459 per year but rises to over MYR 4000 annually in the presence of complications [19]. These figures emphasize the economic impact of diabetes and highlight the need for cost-effective interventions.

Malaysia’s Diabetes Medication Therapy Adherence Clinic (DMTAC) is an ambulatory care initiative where pharmacists collaborate with physicians to improve medication adherence and glycemic control. The program provides diabetic patients with structured follow-up sessions, including medication adherence evaluations, drug-related problem identification, medication counseling, clinical outcome assessments, and diabetes education [20]. Patients enrolled in DMTACs receive additional educational support from pharmacists alongside their usual physician consultations, ensuring comprehensive diabetes management.

Despite the well-documented benefits of pharmacist-led education, there remains a gap in research assessing its impact on clinical outcomes and direct medication costs in Malaysian public hospitals. Unlike European health care systems, where T2DM patients are principally managed in primary care settings and referred to hospitals only for severe complications [21], Malaysian hospitals play a larger role in diabetes care, which includes the specialized diabetes center in each tertiary care hospital where all the diabetic patients can come and take their medications provided by the government [22]. Given the increasing burden of T2DM in Malaysia, it is crucial to evaluate the effectiveness of pharmacist-led educational interventions within the public health care system.

As of right now, no prospective follow-up study has been undertaken at multicenter public hospitals to explicitly investigate the role of pharmacists in diabetic patient management using the DMTAC concept across Malaysia’s public health care system. No research has been undertaken to assess the impact of DMTAC services throughout Malaysia on the treatment expenses generated by public hospitals.

This study aims to investigate the impact of pharmacist-led interventions on disease outcomes and their impact on direct treatment costs within the Malaysian DMTAC program. Specifically, it seeks to determine whether pharmacist-led education and lifestyle modification counseling can enhance glycemic control, reduce unnecessary medication use, and ultimately lower health care expenditures related to diabetes complications. By addressing these aspects, this study will contribute valuable insights into the cost-effectiveness of pharmacist-led interventions in T2DM management within Malaysian public hospitals.

## 2. Methodology

### 2.1. Study Design

A randomized controlled trial was conducted in two tertiary care hospitals in Malaysia to assess the impact of pharmacist-led interventions on diabetes outcomes and direct treatment costs over one year. This study was registered as a multicenter clinical trial on the Australian New Zealand Clinical Trials Registry in compliance with World Health Organization guidelines (ACTRN12621001128886). The direct treatment costs were analyzed at each follow-up with clinical outcomes of disease to determine the economic impact of pharmacist involvement in diabetes management.

### 2.2. Study Population and Study Approvals

This randomized controlled trial (RCT) was conducted at two distinct tertiary care hospitals in Kedah, Malaysia. The approval for conduct and publication was taken from the authorities of the local study center and the Medical Research Ethics Committee under the Ministry of Health Malaysia [KKM/NIHSEC/P18-1307(13)]. Additionally, this study was approved and registered with the National Medical Research Register (NMRR-17-2381-38042).

The sample size for this study was estimated using Power Sample Size Estimation based on the Prior Data technique, referencing a 2016 study by Butt et al. [23]. The calculation aimed to compare the clinical outcomes of the disease at each follow-up and assess the impact on medication costs with pharmacist intervention. To detect a 0.80% difference in HbA1c levels (e.g., 8.48% vs. 9.27%) with 80% power, a 0.05 alpha level, and a standard deviation (σ) of 1.61, approximately 65 participants per group were required. To accommodate a 20% dropout rate, the final sample size was adjusted to 80 diabetic patients per arm. The independent *t*-test was used to test the null hypothesis at a 0.05 significance level. Both relative and absolute measures were taken into account to ensure a comprehensive assessment of the study outcomes.

It is important to note that the pharmacists providing interventions were already employed at the selected study hospitals for routine clinical services; they were not specifically appointed for this study. Therefore, the cost of pharmacist intervention was not calculated, as it was not a primary objective of the current research.

### 2.3. Procedure and Randomization

Patients diagnosed with type 2 diabetes mellitus for at least five years and having an HbA1c level greater than 8.0% were recruited into the study groups based on the determined sample size and following the necessary study approvals. The participants were selected from designated study centers, with 200 diabetic patients enrolled from each hospital. Of these, 100 were assigned to the control group, while the remaining 100 were allocated to the intervention group. Individuals newly diagnosed with type 2 diabetes, pregnant women with diabetes, diabetic patients with HIV or cancer, and patients with incomplete medical records were excluded from this study. A recruitment period of three to four months was established, depending on the influx of patients at the designated hospital. Initially, 600 eligible patients with type 2 diabetes provided consent through an Informed Consent Form and were listed for potential inclusion. Their details were recorded in Microsoft Excel, where a randomization process was conducted to ensure an unbiased distribution into control and intervention groups. To further minimize selection bias, a secondary randomization was performed within each group, ultimately selecting 200 patients per group.

To prevent information contamination between the intervention and control groups, DMTAC services were scheduled exclusively on two selected days per week at all participating hospitals. This protocol ensured that only patients from the intervention group received pharmacist-led educational interventions on those specific days. Additionally, all physicians managing diabetes care for both study groups were well informed about the ongoing research through the Clinical Research Centre of the respective hospitals, thereby eliminating information blindness and potential data contamination. Adult diabetic patients receiving standard care, as per Malaysian Clinical Practice Guideline 2015, at the outpatient diabetic clinics of the designated study hospitals were included in the control group, while those receiving standard outpatient care along with additional educational intervention from pharmacists at the selected hospitals were placed in the intervention group.

The educational interventions provided by DMTAC pharmacists to diabetic patients in all involved hospitals were structured into four comprehensive modules. The first module focused on providing patients with a fundamental understanding of diabetes, including its signs, symptoms, and complications, along with therapeutic goals such as blood glucose targets (FBG and HbA1c). It also included discussions on medication use and adverse effects, self-monitoring of blood glucose, hypoglycemia and hyperglycemia management, proper storage of antidiabetic medications, and addressing patient concerns. The second module emphasized cardiovascular education, covering lipid profiles, blood pressure management, and peripheral vascular disease while explaining the benefits of maintaining target blood glucose levels and the risks of poor glycemic control. Additionally, it provided guidance on the use and side effects of cardiovascular medications, including antihypertensives, antiplatelets, and cholesterol-lowering agents. The third module introduced the benefits of exercise, basic nutritional guidance, and smoking cessation, explaining the health advantages of quitting smoking while reinforcing awareness about hypoglycemic reactions. The fourth and final module addressed diabetes-related complications, including macrovascular and microvascular conditions, with a focus on prevention, recognition, and treatment strategies. It also covered essential foot care education and reiterated the importance of exercise while continuously addressing patient concerns throughout the educational process.

Relevant data were obtained from diabetic clinics for the control group and from DMTAC services for the intervention group at each follow-up. Baseline data collection at the beginning of this study included clinical disease outcomes and medication costs for each diabetic patient. Improvements in clinical outcomes and medication costs were documented individually from patient records during each follow-up visit. To assess changes in direct medication costs relative to baseline observations, two follow-up visits were recorded for both groups. Figure 1 below illustrates the complete study flowchart.

The cost of medication in both of the study groups was calculated at baseline and every follow-up for three months. The cost of medications includes the cost of insulin used, the cost of oral antidiabetics, the cost of medication for diabetic comorbidities and complications, multivitamin and mineral costs, the cost of NSAIDs, and other drugs given to the patients. These other miscellaneous drugs contain other supportive medicines, like omeprazole, bisacodyl, calcium lactate, some antihistamines, and some antiemetics.

### 2.4. Data Collection and Data Management

A pre-validated tool was employed to collect patient information and data in this study. The first instrument used was a data collection form, which included patient demographic information and disease-specific clinical outcomes such as the fasting blood glucose (FBS), glycated hemoglobin (HbA1c), blood pressure, and lipid profile status of individuals with diabetes. The medication costs were obtained from the formulary of the selected study hospitals, and the total cost was computed for a six-month duration based on the daily dosage and treatment regimen. During the study period, patient medical hospital files were accessed for all included diabetic patients who agreed to participate by providing written consent, and their disease and treatment outcome information was recorded using the validated data collection form. The hospital authorities graciously provided the medication price list with the consent of the hospital directors.

All the relevant patient information was collected from hospital medical files at every follow-up, allowing for continuous assessment of improvements in both study groups, which were compared at each follow-up visit.

### 2.5. Validation of Tools for Data Collection

The data collection form’s content validity was evaluated through a panel of specialists, comprising 2 specialists from the diabetes clinics and 2 chief pharmacists from both study locations. The main instrument for data collection was distributed to clinical specialists for assessment of the relevance, representation, and suitability of all the sections of the data collection tool. Each specialist recommendation was incorporated into the final data collection tool.

Face validation involved delivering the data collection instrument to a group of 15 individuals who evaluated its clarity and comprehensibility. The face validation group consisted of five endocrinologists from each study location, five specialists from an established public institution, and five faculty members from the Pharmacy Faculty at a private medical university within Kedah, Malaysia. They were asked to evaluate aspects, including readability, clarity, language, and typographical problems. The instrument for data collection was completed following a sequence of discussions with these specialists.

### 2.6. Data Requisite for Study Tool

Throughout the trial period, the medical records of all the participating diabetic patients who provided written consent were reviewed, and the data collection form was used to record their disease and treatment outcome. The data collection instrument intended for the present study was segmented into 3 sections.

The data collection form was divided into three sections. The first section gathered demographic information, including patient characteristics, social factors, and medical and medication history. The second section focused on clinical outcomes, documenting laboratory values, improvements in signs and symptoms, and progression of diabetic complications at each follow-up visit. The third section documented medication costs, including expenses for diabetes treatment; management of comorbidities; and costs associated with diabetic complications, such as diabetic neuropathy, nephropathy, retinopathy, diabetic foot, and vasculopathy. All the data were extracted from patients’ hospital medical records during the follow-up visits.

The primary objective of this study was to evaluate the impact of pharmacist intervention on medication costs while assessing improvements in disease outcomes. These outcomes included fasting blood glucose (FBS), glycated hemoglobin (HbA1c), blood pressure, and lipid profile status in individuals with diabetes. Improvements in both study groups were documented and analyzed at each follow-up visit.

### 2.7. Statistical Analysis

The data were analyzed using SPSS version 24. Continuous data were presented as the mean ± standard deviation (SD), while categorical data were expressed as counts and percentages (N%). The normality of the data was assessed using skewness, kurtosis, the Shapiro–Wilk test, and the Kolmogorov–Smirnov (KS) test. If the data followed a normal distribution, an independent *t*-test was applied to evaluate the null hypothesis. Partial Eta Squared (η^2^) was used in the initial analysis to determine the effect size in a one-way ANOVA, classified by Cohen as minor (0.01 ≤ η^2^ ≤ 0.06), medium (0.06 ≤ η^2^ ≤ 0.14), or large (η^2^ ≥ 0.14). The statistical significance was set at a *p*-value of less than 0.05. Independent *t*-tests and paired *t*-tests were used to determine the *p*-values for clinical and cost-related variables, respectively, using this significance level. Key terms include SD (standard deviation) and df (Degrees of freedom).

### 2.8. Confidentiality and Data Security

All the subject identities were recorded in an encrypted database and were exclusively linked to a study-identifying number for this investigation. Identification numbers were utilized in place of patient identifiers on the subject datasheets. The entire dataset was entered into a computer that was encrypted. The data from the computer were transmitted to a USB flash drive after this study was finished, and the computer’s data were subsequently removed. In compliance with the requirements of MREC Malaysia, investigators will retain USB flash drives and any hardcopy data for at least three years following the conclusion of this study. The USB flash drive and its contents will be entirely erased following the storage period. The participants were not allowed to access their individual data since it would be combined into a database. The patients were allowed to communicate directly with the investigators in order to request access to this study’s findings.

## 3. Results

A total of 400 diabetic adult patients were enrolled at baseline. Of these, 299 patients completed two follow-up visits over the one-year study duration. At baseline, a statistical analysis was conducted on all the demographic variables of the included patients to identify variations during the recruitment process, thereby reducing the likelihood of bias in the study.

The study groups’ baseline characteristics, as illustrated in Table 1, indicate that no statistically significant differences (*p* > 0.05) were found in the demographics of the patients who were included. Furthermore, the intervention group and the control group did not exhibit any significant differences in their educational backgrounds. On the other hand, there were statistically significant differences in subject age and duration of diabetes between the two groups.

### 3.1. Clinical Outcome Measurements

This study evaluated changes in glycemic control under random blood glucose levels (RBS) and fasting blood glucose levels (FBS), alongside systolic (BP systolic) and diastolic (BP diastolic) blood pressure, glycated hemoglobin (HbA1c), total cholesterol, triglycerides, and High-Density Lipoprotein Cholesterol (HDL-C) and Low-Density Lipoprotein Cholesterol (LDL-C).

The baseline did not reveal any significant differences between the control group (CG) and the intervention group (IG). The alterations in the aforementioned parameters of the patients are illustrated in Table 2 and Table 3 at six-month intervals (follow-up 1) and one year from baseline (follow-up 2). The changes observed in the control and intervention groups are illustrated in Table 2 and Table 3. The results of this study indicate that the average reduction in glycated hemoglobin (HbA1c) from the beginning to the end of this study was approximately 1.43% in the control group. In contrast, the intervention group, where pharmacists were actively involved, experienced a significantly greater reduction of 2.82%. Additionally, the IG subjects exhibited a statistically significant difference between follow-ups 1 and 2 in blood sugar, blood pressure, triglyceride, and LDL-C levels, whereas no such significant difference was observed in the CG subjects.

### 3.2. Cost of Medication Measure

The statistical significance of all the characteristics that influence the cost of medication was assessed at baseline to identify any discrepancies. The differences were found to be statistically insignificant. Consequently, it was demonstrated that there was no credible distinction between group characteristics that could potentially influence the cost of medication. Table 4 illustrates each statistical difference. The medication cost for diabetic patients in both study arms was determined for three months using baseline data, as the hospitals issued the medicines for a specific period. Table 4 contains the cost of medication for both study groups. All these baseline costs were comparable between the control and intervention groups.

Insulins were available in the selected study hospitals in three different types, each with a varied duration of action. Nevertheless, the patients at baseline were only prescribed intermediate- and long-acting insulins. These insulins were also priced differently; the cost of long-acting insulin was MYR 25.66 per 100 units, while intermediate-acting insulin was priced at MYR 31.55 per 100 units. In a similar way, oral antidiabetics’ costs fluctuated in accordance with the medications that were available. The single-active drug tablet was less expensive; however, the combination of two medications in a single tablet was more costly.

The cost of medication for the control group over a three-month period was MYR 763.90 at baseline, whereas the intervention group’s expense was MYR 783.01. The intervention group had a significantly greater number of patients with intermediate-acting insulin than the control group, which resulted in the slight variation. However, this difference was incidental, as all patients were randomly assigned to their respective groups. Intermediate-acting insulin was priced higher than long-acting insulin. In a similar way, the prices of all available medications varied.

Moreover, a small variation was observed in both study groups for diabetic complication medication. The intervention group had a mean cost of MYR 347.49, whereas the control group had an average price of MYR 350.21. Nevertheless, the costs of diabetic complications lacked a statistically significant difference (*p* = 0.253) between the two groups.

Likewise, a statistically insignificant correlation was seen in both study groups (*p* = 0.150) in other miscellaneous drugs. Additional supportive medications, including calcium lactate, bisacodyl, omeprazole, and certain antihistamines and antiemetics, are present in this miscellaneous drug category.

Medication costs in the intervention group continued to decrease significantly at the first follow-up as a result of lifestyle modification and pharmacist intervention. On the other hand, in the control group, there was little variation in the cost of medication. The cost of insulin increased as a result of the change in treatment of some patients who were not showing any response to a switch from the long-acting to intermediate-acting variety.

Intermediate-acting insulin costs approximately MYR 31.55 per 100 units, while long-acting insulin costs approximately MYR 25.66 per 100 units. Furthermore, some patients in the control group were shifted to a combination medication of oral antidiabetics and insulin when they were not responding to the prescribed pharmaceutical treatment.

Insulin consumption costs were significantly different between the two groups during the first follow-up (*p* = 0.009). The costs of oral antidiabetic medicines were additionally considerably greater in the control group compared to the intervention group (*p* ≤ 0.001). In the intervention, the group’s use of Nonsteroidal Anti-Inflammatory Drugs (NSAIDs) and other medications declined during follow-up 1. However, the control group of the present study experienced a rise in their consumption during the same period. Table 5 demonstrates these changes.

There was a statistically significant difference between the two study groups at the second follow-up regarding the medication cost of all stated variables. In the control group, some patients had to change their medication regimen since their bodies were not responding to the insulin they had previously been given. Some of them were able to increase the daily unit dosages of insulin. Conversely, physicians in both study locations transitioned a portion of the participants to intermediate-acting insulins. The cost of medication per patient will ultimately increase if the unit dose is increased or the insulin is changed.

In both study groups, there was a statistically significant increase in the cost of insulin per patient (*p* ≤ 0.001), as illustrated in Table 6. The control group spent an average of MYR 918.62 on insulin for each patient, while the mean cost of insulin per patient in the intervention group was approximately MYR 811.94. That differential was due to the fact that the cost increase per patient over baseline in the control group was approximately MYR 154.72. The intervention group, on the other hand, had a far lower cost increase of only MYR 28.93 per patient.

The control group had an increase from baseline in the cost of multivitamins and minerals during the follow-up period, resulting in a cost of MYR 2.59 per patient. However, the intervention group experienced a decrease of up to MYR 0.37 per patient from the current study’s baseline data. The same change has been found for Nonsteroidal Anti-Inflammatory Drugs as well as other supportive medications, such as bisacodyl, omeprazole, calcium lactate, certain antihistamines, and antiemetics.

The modifications are illustrated in Table 6 as follows:

**Table 6 healthcare-13-00901-t006:** Cost of medication for patients at follow-up 2 for three months.

Parameters	Cost for 3 Months Mean ± SD CG (N = 143)	Difference from Baseline Price	Cost for 3 Months Mean ± SD IG (N = 156)	Difference from Baseline Price	*t*-Statistic (df)	*p*-Value
Insulin	MYR 918.62 ± 230.43	MYR +154.72	MYR 811.94 ± 165.69	MYR +28.93	4.62 (1, 297)	<0.001
Oral antidiabetics	MYR 87.03 ± 52.43	MYR +38.32	MYR 56.37 ± 27.67	MYR +7.62	6.39 (1, 297)	<0.001
Diabetic comorbidities	MYR 85.20 ± 10.26	MYR +6.4	MYR 80.75 ± 6.05	MYR +1.87	4.60 (1, 297)	<0.001
Diabetic complications	MYR 379.84 ± 39.80	MYR +29.63	MYR 359.37 ± 33.21	MYR +11.88	4.84 (1, 297)	<0.001
Multivitamins and minerals	MYR 18.44 ± 4.77	MYR +2.59	MYR 14.88 ± 2.06	MYR −0.37	8.47 (1, 297)	<0.001
NSAIDs	MYR 3.23 ± 0.97	MYR +0.55	MYR 2.21 ± 0.80	MYR −0.38	9.90 (1, 297)	<0.001
Others	MYR 26.72 ± 6.48	MYR +3.86	MYR 19.88 ± 3.73	MYR −2.22	11.29 (1, 297)	<0.001

*p*-value between both study groups after 3–4 months of follow-up 3. Independent Samples *t* Test. The *t*-statistic is farther from zero than the critical values considered, which rejects the null hypothesis. C.I. = Confidence Interval (Group Statistics); SD = standard deviation; and df = Degrees of freedom.

## 4. Discussion

The present study was the first of its kind in Malaysia to investigate pharmacists’ contributions to medication cost reduction. It was conducted over a year in two distinct tertiary care hospitals. Only a few studies have attempted to assess the full cost of diabetes mellitus therapy in Malaysia. Nevertheless, not a single one of these studies observed how dietary or educational modifications affected the efficacy of diabetes medications.

Two parallel arms, a control and intervention group, formed the current investigation. Patients received the standard/usual treatment in the control group, whereas the pharmacist-led educational intervention was given to the intervention group. The Diabetes Medication Therapy Adherence Clinic (DMTAC) pharmacist provided continuous patient education to diabetic patients in the intervention group, covering topics such as disease management, diet modification, lifestyle modifications, physical activities, and treatment adherence. The consumption and utilization of drugs can be reduced with the assistance of such educational interventions. The Malaysian Health Ministry’s objective of establishing DMTAC departments in all public tertiary care institutions was to reduce the cost of treatment [20].

### 4.1. Impact on Glycated Hemoglobin (HbA1c)

The results of this study indicate that the average reduction in glycated hemoglobin (HbA1c) from the beginning to the end of the study was approximately 1.43% in the control group. In contrast, the intervention group, where pharmacists were actively involved, experienced a significantly greater reduction of 2.82%. This demonstrates a statistically significant improvement (*p* < 0.001) in the intervention group compared to the control group. The overall decline in HbA1c observed in both groups can be attributed to the involvement of specialized health care providers in tertiary care hospitals across Malaysia. However, the substantially greater reduction in HbA1c among patients in the intervention group highlights the impact of pharmacist-led collaborative care.

These findings align with a randomized controlled study conducted by Lim et al., which reported a significant decrease in HbA1c, with an average reduction of 0.91% in the intervention group and 0.08% in the control group (*p* ≤ 0.011) [24]. Similarly, a randomized controlled trial by Butt et al., conducted at Universiti Kebangsaan Malaysia Medical Centre (UKMMC), found a significant reduction in HbA1c from 9.677% to 8.48% (*p* ≤ 0.001) in the intervention group [23]. In contrast, the control group showed no statistically significant reduction (9.64% to 9.26%; *p* = 0.14). A systematic review by Iqbal et al. supports these findings, suggesting that pharmacist-led collaborative care can lower HbA1c levels by an average of 0.75% [5].

While the present study’s results are consistent with prior research, they differ slightly from those of a six-month randomized controlled trial by Jarab et al., where HbA1c in the intervention group decreased by 0.8%, whereas it slightly increased in the control group [25]. Similarly, Phumipamorn et al. reported that a six-month pharmacist-led intervention resulted in an average HbA1c reduction of 0.8% in the intervention group, while the reduction in the control group was not statistically significant [26]. Their study’s structured pharmacist-led educational strategy, which focuses on individualized patient counseling, medication adherence techniques, and lifestyle changes, may have contributed to the intervention group’s higher HbA1c reduction.

Additionally, retrospective studies have reported similar outcomes. A study by Abdullah et al. found that pharmacist-led collaborative care improved HbA1c levels by an average of 1.32% [27]. Meanwhile, research by You et al. showed an average HbA1c reduction of 1.0%, with a standard deviation of ±1.7%, due to pharmacist involvement in diabetes management [28]. These findings collectively reinforce the effectiveness of pharmacist-led interventions in improving glycemic control among diabetic patients.

### 4.2. Medication Costs and Analysis

Statistical significance was assessed for all characteristics that influenced the cost of medication at the baseline to identify variances among the two study groups. In both study cohorts, no statistically significant differences were observed in the factors that could affect medication expenses. At baseline and at each three-month follow-up, the medication expenses for both groups were calculated, as the hospitals were only administering medications for a three-month period. The expense of medications includes the cost of insulin, oral antidiabetics, medications for complications and diabetic comorbidities, and minerals and multivitamins, as well as Nonsteroidal Anti-Inflammatory Drugs and other drugs administered to patients. Further various medications include medicinal agents, such as omeprazole, bisacodyl, calcium lactate, some antihistamines, and certain antiemetics.

At the beginning of this study, the medication costs were comparable between both groups. The analysis considered the expenses associated with prescribed medications in each group. The group receiving the intervention exhibited a slight variation, as it contained more patients utilizing intermediate-acting insulin than the control group. The cost of intermediate-acting insulin was greater than that of long-acting insulin. The prices of various insulins varied; long-acting insulin was priced at MYR 25.66 per 100 units, while intermediate-acting insulin was priced at MYR 31.55 per 100 units. The expense of oral antidiabetics fluctuated based on the available medication list. The individual active medicine tablets were less expensive; however, the combination tablet containing two pharmaceuticals was costly [29].

This study also analyzed the costs associated with treating diabetic complications, considering the expenses for each medication used. However, no statistically significant association (*p* = 0.253) was observed between the costs of diabetic complications in either study group. A non-significant statistical association (*p* = 0.150) was detected in both study groups regarding other miscellaneous medications.

This study also examined changes in insulin costs between the control and intervention groups over the follow-up period. A statistically significant variation (*p* < 0.001) in the cost of insulin was noted during the second follow-up among the study groups. The expense of insulin in the control group underwent an additional increase. The potential cause was the alteration in insulin. The alteration in insulin directly increased the medication expenses as specific people unresponsive to long-acting insulin therapy were transitioned to intermediate-acting insulin in the control group [30]. The long-acting insulin price was approximately MYR 25.66 per 100 units, whereas the intermediate-acting insulin price was MYR 31.55 per 100 units.

The mean prescription expense for antidiabetics (oral) in the control group increased from the baseline by approximately MYR 17.03. However, the rise in the intervention group was around MYR 2.78 per patient. The expense of oral antidiabetic medications was markedly (*p* ≤ 0.001) elevated in the control group compared to the intervention group at follow-up 2. The likely explanation for this differential might be that the participants in the control group who showed a lack of responsiveness to the prescribed oral antidiabetic pharmacotherapy combined with the insulin were switched to a combined regimen of oral antidiabetics. The price of a single, active drug was shown to be less than that of a combination of two or three active compounds in the same tablet formulation; for instance, the price of a 500 mg metformin tablet was MYR 0.10 each. In contrast, the price of metformin 1 g combined with sitagliptin 50 mg was 1.88 per tablet.

Utilization of NSAIDs and other medications persistently declined during the second follow-up for the intervention group, while it grew in the control group. The decline in the intervention group may be attributed to lifestyle changes, physical activity, and dietary modifications, alongside pharmacist educational intervention. However, in the control group, the pharmaceutical costs exhibited minor fluctuations. Drug cost increased in both of the study groups during the second follow-up.

In the current study, the rise in medication costs in the control group was thrice that of the intervention group when taking into account insulin and oral antidiabetics. The results of the second follow-up of our investigation aligned with a study conducted in India by Dussa and colleagues, which indicated that pharmacist involvement reduced hospitalization costs for diabetic patients [31]. The reported study’s deficiency was its single-center design, which involved only one population undergoing intervention. Our investigation included two distinct patient groups.

At follow-up 2, a statistically significant difference (*p* < 0.001) was observed between the two study groups for all included cost parameters in the present investigation. A statistically significant difference (*p* ≤ 0.001) in insulin expenses per patient was seen between the two study groups. The average price of insulin in the control group was MYR 918.62, whereas the average price per patient in the intervention group was roughly MYR 811.94. The increase in price is evinced by the increase per patient from baseline being around MYR 154.72 in the control group, whereas in the intervention group, the increase per patient was only MYR 28.93 from the baseline. Likewise, the cost of multivitamins and minerals increased from baseline in the last follow-up, reaching MYR 2.59 per patient in the control group. In the intervention group, it decreased by as much as MYR 0.37 per patient relative to the baseline. Identical variations have been noted in NSAIDs and other adjunctive medications, such as omeprazole, bisacodyl, calcium lactate, some antihistamines, and specific antiemetics. The possible explanation for the variance or rise in drug costs in the control group could be similar to that of insulin. In the control group, certain individuals who were not responsive to the insulin that was prescribed were required to change the pharmacological regimen. For example, several individuals increased their daily unit dosages of insulin.

On the other hand, physicians in both study locations advised a switch for some people to intermediate-acting insulins. An increase in the unit dose or a modification in the type of insulin will ultimately elevate the prescription cost per patient. A similar rationale could be behind an increase in the prices of other medications. Following pharmacist intervention, the intake of additional oral antidiabetics, multivitamins and minerals, NSAIDs, and other medications was limited during follow-up 2 for the intervention group, subsequent to consultations between DMTAC pharmacists and physicians, as patients in the intervention group adopted a more balanced diet due to the pharmacist-led educational initiatives provided by the DMTAC.

The prescription costs in the intervention group included in the DMTAC program were markedly lower than those in the control group of the present study. The only difference between the two groups was the extent to which pharmacists were involved in the intervention group; otherwise, both groups used the same health care facilities and physicians. The participating pharmacists in Malaysia are not compensated for their DMTAC services. This service provides patients with extra value. Indeed, the principal objective of DMTAC services in Malaysia was to reduce the cost of diabetes mellitus treatment in the country [20]. In other words, it may be stated that the Government can achieve significant savings without paying additional expenses from the Ministry through the engagement of pharmacists. Everyone in Malaysia with type 2 diabetes should be able to benefit from this service, not limited to particularly few individuals. It should be available at all hospitals in the country.

### 4.3. Comparison with Previous Studies

The research to date regarding the cost-effectiveness of educational interventions by pharmacists has been inadequate. The cost-effectiveness of diabetic medical therapy assessment and care DMTAC services and pharmacist interventions in Malaysian tertiary care facilities deserve greater attention. Our study shows that when pharmacists are not involved, an additional MYR 236.07 per patient was spent by the second follow-up in comparison to the baseline. However, the additional cost per patient was only MYR 47.33 when a pharmacist was involved. The cost variation was considerable between the two study groups. The results of our study align with those of research conducted in New Jersey to assess the pharmacist-led MTDM program [32]. The study by Maeng et al. indicates that the within a year, the MTDM program reduced hospitalization rates by as much as 19.6 percent while also bringing down average health care expenses per patient by around 13%.

Likewise, multiple investigations have demonstrated that management of medication with a clinical pharmacist leads to enhanced clinical outcomes and reduced overall drug costs [33,34,35]. Cranor and colleagues conducted a pre−post cohort analysis revealing that pharmacist assistance throughout seven follow-ups resulted in a reduction in total average direct medical costs by around USD 1200 to USD 1872 per patient over one year compared to baseline [33]. All the aforementioned studies have shown a beneficial impact of pharmacist intervention in reducing prescription costs for diabetic patients and the health care system overall. The positive impact that pharmacists have on the health care systems of a variety of countries worldwide is evidenced by the interventions that they provide in each of the studies discussed above. Nevertheless, pharmacist intervention is not standardized globally. Pharmacist intervention programs are unique to each country and contain a variety of unique elements. It is challenging to determine which component of the intervention program had the greatest impact on the desired outcomes. Therefore, more investigations are required to establish the most effective and feasible approach by correlating the impact of the various elements during educational interventions on clinical outcome measures.

This randomized controlled trial was one of the pioneering studies conducted in two tertiary care hospitals in Malaysia to evaluate the role of pharmacists in the Medication Therapy Adherence Clinic (MTAC) program for diabetes mellitus. This study assessed the impact of pharmacist intervention on clinical outcomes and medication costs for diabetic patients. While the one-year follow-up provided valuable insights, a longer follow-up would offer a clearer understanding of disease progression, particularly for diabetic complications. Since this study was conducted in only one state of Malaysia, its findings cannot be generalized to the entire country. Therefore, similar studies should be conducted across all states to assess the overall impact of the DMTAC program.

Based on the findings, several recommendations can enhance future research and implementation. The Health Ministry should conduct a nationwide, long-term follow-up study for a more comprehensive evaluation. Given the demonstrated effectiveness of pharmacist involvement in diabetes management, the government should consider expanding DMTAC services to all hospitals and clinics across Malaysia. Additionally, the impact of pharmacist interventions should be explored for other chronic conditions, such as asthma, arthritis, cancer, Alzheimer’s disease, and other dementias. Lastly, DMTAC services should also be introduced in private hospitals and clinics to ensure wider accessibility and effectiveness.

## 5. Conclusions

In general, health care professionals are at the front line to provide compliance therapy for diabetes mellitus according to the Clinical Practice Guidelines (CPGs) in the health care centers recruited for this study for both study groups. Yet the participation of pharmacists in the form of interventions results in improved control of clinical outcomes of diabetes mellitus. This study’s results confirm that DMTAC is an effective program in Malaysia. Our study suggests that in general diabetic patients, including patients with uncontrolled diabetes, should be involved in pharmacist-led interventional programs to prevent further clinical complications and enhance their health-related quality of life.

## Figures and Tables

**Figure 1 healthcare-13-00901-f001:**
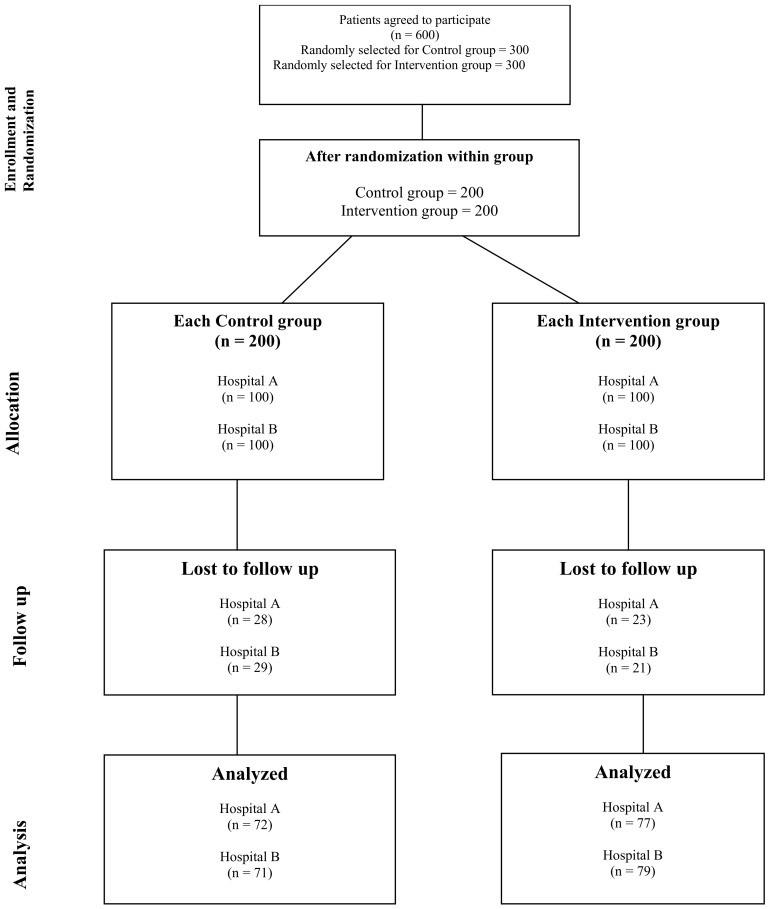
Flowchart of this study.

**Table 1 healthcare-13-00901-t001:** Baseline demographic characteristics + clinical variables of study subjects.

Variables		Frequency		*p*-Value
		** CG n (%)	** IG n (%)	
Hospital Name				0.819 *
	Hospital A	71 (48.6)	75 (51.4)	
	Hospital B	70 (47.3)	78 (52.7)	
Gender				0.136 *
	Male	74 (52.5)	67 (47.5)	
	Female	67 (43.8)	86 (56.2)	
Ethnicity				0.400 *
	Malay	104 (46.0)	122 (54.0)	
	Chinese	25 (52.1)	23 (47.9)	
	Indian	12 (60.0)	8 (40.0)	
Age (mean, SD)				<0.001 ^#^
	-	58.68 ± 6.08 N = 141	61.63 ± 6.17 N = 153	
Duration of diabetes (years, SD)				0.032 ^#^
	-	9.62 ± 2.35	10.24 ± 3.23	
Disease outcomes				
	HbA1c (%)	11.15 ± 1.33	11.68 ± 1.50	0.061
	FBS (mmol/L)	14.63 ± 1.38	14.44 ± 1.53	0.265
Residence status				0.438 *
	Urban	66 (46.2)	77 (53.8)	
	Rural	75 (49.7)	76 (50.3)	
Employment status				0.587 *
	Unemployed	67 (46.2)	78 (53.8)	
	Employed	74 (49.0)	77 (51.0)	
Type of daily diet				0.328 *
	Vegetarian	79 (50.6)	77 (49.4)	
	Non-vegetarian	62 (44.9)	76 (55.1)	
Smoking status				0.635 *
	Yes	26 (51.0)	25 (49.0)	
	No	115 (47.3)	128 (52.7)	
Exercise status				0.178 *
	Yes	41 (54.7)	34 (45.3)	
	No	100 (45.7)	119 (54.3)	
Type of antidiabetic therapy				0.882 *
	Oral only	82 (48.8)	86 (51.2)	
	Insulin	44 (47.8)	48 (52.2)	
	Oral + insulin	15 (44.1)	19 (55.9)	

* Chi-square test; ^#^ independent *t*-test. ** CG = control group; IG = intervention group.

**Table 2 healthcare-13-00901-t002:** Differences in outcomes in the control group (N = 143).

Outcome Measure	Baseline	Follow-Up 1 (After 6 Months of Baseline)			Follow-Up 2 (After 1 Year of Baseline)		
		Mean ± SD	Mean Difference	*p*-Value ^α^	Mean ± SD	Mean Difference	*p*-Value ^β^
FBS (mmol/L)	14.63 ± 1.37	12.09 ± 1.34	−2.54	<0.001	11.79 ± 1.33	−0.30	0.067
RBS (mmol/L)	17.65 ± 1.34	15.93 ± 1.35	−1.72	<0.001	16.81 ± 1.39	+0.88	0.347
HbA1c (%)	11.15 ± 1.32	10.10 ± 1.04	−1.05	<0.001	9.72 ± 1.02	−0.38	0.003
BP Systolic (mmHg)	132.71 ± 6.09	131.63 ± 5.98	−0.08	0.365	135.13 ± 5.99	+0.58	0.276
BP Diastolic (mmHg)	84.37 ± 5.63	85.17 ± 5.05	+0.80	0.455	84.24 ± 5.49	−0.93	0.064
Total Cholesterol (mmol/L)	5.67 ± 0.35	5.43 ± 0.29	−0.24	0.004	5.41 ± 0.30	−0.002	0.457
Triglyceride (mmol/L)	1.88 ± 0.18	1.70 ± 0.14	−0.18	0.001	1.76 ± 0.12	−0.06	0.345
LDL-C (mmol/L)	2.81 ± 0.17	2.73 ± 0.12	−0.08	0.352	2.74 ± 0.12	+0.01	0.542
HDL-C (mmol/L)	1.04 ± 0.02	1.17 ± 0.03	+0.13	0.001	1.16 ± 0.03	−0.01	0.067

Paired *t*-test was used to find the *p*-values. *p*-value ^α^ = *p*-value between baseline × after 6 months (follow-up 1). *p*-value ^β^ = *p*-value between follow-up 1 × after 6 months.

**Table 3 healthcare-13-00901-t003:** Differences in outcomes in the intervention group (N = 156).

Outcome Measure	Baseline	Follow-Up 1 (After 6 Months of Baseline)			Follow-Up 2 (After 6 Months of 1st Follow-Up)		
		Mean ± SD	Mean Difference	*p-*Value ^α^	Mean ± SD	Mean Difference	*p*-Value ^β^
FBS (mmol/L)	14.44 ± 1.53	10.01 ± 1.51	−4.43	<0.001	7.63 ± 1.45	−2.38	<0.001
RBS (mmol/L)	18.32 ± 1.48	13.87 ± 1.45	−4.45	<0.001	11.54 ± 1.39	−2.33	<0.001
HbA1c (%)	11.69 ± 1.50	9.94 ± 0.92	−1.75	<0.001	8.87 ± 0.79	−1.07	<0.001
BP Systolic (mmHg)	136.87 ± 5.60	131.27 ± 5.49	−5.60	<0.001	129.97 ± 5.31	−1.30	<0.001
BP Diastolic (mmHg)	85.49 ± 5.94	83.45 ± 5.87	−2.04	0.001	81.31 ± 5.05	−2.14	0.001
Total Cholesterol (mmol/L)	6.16 ± 0.35	5.41 ± 0.32	−0.75	<0.001	5.34 ± 0.0.29	−0.07	0.067
Triglyceride (mmol/L)	1.97 ± 0.22	1.82 ± 0.22	−0.15	0.001	1.71 ± 0.14	−0.11	0.003
LDL-C (mmol/L)	3.45 ± 0.20	3.01 ± 0.35	−0.44	<0.001	2.72 ± 0.10	−0.29	<0.001
HDL-C (mmol/L)	0.91 ± 0.04	1.01 ± 0.10	+0.1	0.001	1.02 ± 0.11	+0.01	0.568

Paired *t*-test was used to find the *p*-values. *p*-value ^α^ = *p*-value between baseline × after 6 months (follow-up 1). *p*-value ^β^ = *p*-value between follow-up 1× after 6 months.

**Table 4 healthcare-13-00901-t004:** The cost of medication in Malaysian Ringgit (MYR) for patients at baseline for 3 months.

Parameters	Cost in MYR for 3 Months Mean ± SD ** CG (N = 143)	Cost in MYR for 3 Months Mean ± SD ** IG (N = 156)	*t*-Statistic (df)	*p*-Value
Insulin	763.90 ± 140.74	783.01 ± 149.67	−1.13 (1, 297)	0.258
Oral antidiabetics	48.71 ± 6.52	48.75 ± 6.94	−0.04 (1, 297)	0.963
Diabetic comorbidities	78.80 ± 7.55	78.88 ± 2.97	0.91 (1, 297)	0.908
Diabetic complications	350.21 ± 21.57	347.49 ± 19.58	1.14 (1, 297)	0.253
Multivitamins and minerals	15.85 ± 4.20	15.25 ± 3.22	1.38 (1, 297)	0.167
NSAIDs	2.68 ± 0.73	2.59 ± 0.55	1.25 (1, 297)	0.211
Other medications	22.86 ± 5.09	22.10 ± 3.96	1.44 (1, 297)	0.150

** CG = control group; IG = intervention group. *p*-value between both study groups. SD = standard deviation; df = Degrees of freedom. Independent Samples *t* Test. Note: 1 Malaysian Ringgit (MYR) is approximately equal to 0.223 US Dollars (USD).

**Table 5 healthcare-13-00901-t005:** Medication costs for patients during the first follow-up for a period of three months.

Parameters	Cost for 3 Months Mean ± SD CG (N = 143)	Difference from Baseline Price	Cost for 3 Months Mean ± SD IG (N = 156)	Difference from Baseline Price	*t-*Statistic (df)	*p*-Value
Insulin	RM 874.43 ± 218.33	RM +110.53	RM 816.40 ± 163.91	RM +33.39	2.61 (1, 297)	0.009
Oral antidiabetics	RM 65.74 ± 35.50	RM +17.03	RM 51.53 ± 12.68	RM +2.78	4.68 (1, 297)	<0.001
Diabetic comorbidities	RM 81.79 ± 8.91	RM +2.99	RM 80.16 ± 5.73	RM +1.28	1.88 (1, 297)	0.060
Diabetic complications	RM 361.32 ± 32.13	RM +11.11	RM 350.40 ± 23.59	RM +2.91	3.36 (1, 297)	0.001
Multivitamins and minerals	RM 16.80 ± 4.60	RM +0.95	RM 15.38 ± 3.32	RM +0.13	3.06 (1, 297)	0.002
NSAIDs	RM 2.85 ± 0.79	RM +0.17	RM 2.32 ± 0.75	RM −0.27	5.91 (1, 297)	<0.001
Other medications	RM 24.34 ± 6.11	RM +1.48	RM 21.15 ± 4.23	RM −0.95	5.27 (1, 297)	<0.001

*p*-value between both study groups after 3–4 months of baseline. Independent Samples *t* Test. The *t*-statistic is farther from zero than the critical values considered, which rejects the null hypothesis. C.I. = Confidence Interval (Group Statistics), SD = standard deviation, and df = Degrees of freedom.

## Data Availability

Data are available upon reasonable request.

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
