# Peer review of "Impact of Pharmacist Educational Intervention on Costs of Medication with Improved Clinical Outcomes for Diabetic Patients in Various Tertiary Care Hospitals in Malaysia: A Randomized Controlled Trial"

_healthcare, 2025, doi:10.3390/healthcare13080901_

Round 1
Reviewer 1 Report
Comments and Suggestions for Authors
I like the study, but think that the manuscript has to be rewritten and that the analysis can be done better. Please find some suggestions.

Author Response
Comments 1: Title • I recommend to put the clinical outcomes also into the title. • It would be good to make clear that the education is for the patients by the pharmacists in the abstract and throughout the document.
|
Response 1: Thank you for your valuable feedback. We agree with your suggestion and have accordingly revised the title. The updated title, along with the changes, has been highlighted in red for your reference. |
Comments 2: Abstract • I am not familiar with the currency 'RM'. Please write it out the first time. • There were two public hospitals how were patient numbers included for intervention and control group distributed over these locations? Evenly? • From the abstract I do not understand the results on treatment expenses for differences between the group. What costs were taken into account? Only medication costs - was medication then stopped in the intervention group? Costs on additional counselling? |
Response 2: Agree. We have, accordingly, changed and modified. Please refer to the updated abstract. The changes made are highlighted in red.
Comments 3: Introduction • The introduction may be rewritten to better introduce the research question by a logical flow: e.g. disease frequency and meaning for patient lives, costs for society, health professionals impact in general and pharmacists' impact on patient understanding of disease. Please be clear on what the relation of lifestyle modification is on treatment costs and outcomes. Please state why cost-effectiveness in medication treatment is so meaningful (being more adherent means more dispenses and more costs for drug treatment!). It seems that it is more important to use medication and lifestyle improvement effectively to prevent complications and hospital admissions? The paragraph to this might be written more clearly to what costs are spent on prevention, on treatment of the disease and on treatment of complications-costs on medication used and costs for services provided. Please also give some information on the specific situation in Malaysia. In European countries type 2 diabetes patients would be treated in primary care and only for severe complications in hospitals. Objective: • In the introduction it is stressed that collaborative care from several healthcare practitioners is needed. It is not clear why then this study focusses on an intervention from pharmacists -what does 'governed' mean? • The objective is on a 'correlation' do you mean 'association'? And why not estimate the effect of an intervention by using regression analysis? • Please state the outcomes specifically in the objective: what clinical outcomes are taken into account? What treatment costs are included (see above: drugs and intervention costs)? Response 3: Thank you for your valuable comments and insightful suggestions. We greatly appreciate your time and effort in reviewing our manuscript. As per your recommendations, we have made significant revisions to the Introduction and Objective sections to enhance clarity and logical flow. Major changes have been incorporated to better introduce the research question, highlight the role of pharmacists in diabetes management, clarify cost-effectiveness considerations, and specify the clinical outcomes and treatment costs included in the study. Additionally, we have provided relevant details on the healthcare system in Malaysia for better context. All the requested changes have been made and are highlighted in red for your reference. Please refer to the updated file for the revised version..
Comments 4: Methodology
(This is referred to in the methods section 2.2- however, I do not understand how you can adjust for baseline observations by an ANOVA. It is said that 'changes relative to baseline observations' were calculated. I propose not to use relative but absolute measures - and from table 3.2 in the results it can be seen that absolute values were taken and differences between follow up and baseline were calculated - this is nice and should be clearly described in the methods.)
Please also describe what standard care in Malaysia means with regard to the outcome measures and pharmacist engagement.
Were patients blinded? Pharmacists could not be blinded - but were other healthcare providers (e.g. those who assessed clinical parameters)?
Response 4: Thank you for your detailed and insightful comments. We have carefully incorporated all the suggested revisions into the Methodology section to enhance clarity and accuracy. The study design, sample size calculation, intervention details, patient inclusion criteria, data collection methods, and statistical analysis have been revised accordingly. Additionally, we have addressed concerns regarding effect estimation, cost analysis, and blinding procedures.
All changes have been made in the revised version and are highlighted in red for your reference. Please refer to the updated file.
Comments 5: Results
Table 3.4 could be part of the baseline table 3.1 When performing regression analysis, there could be one table for the baseline parameters and one for the effect measures.
Response 5: Thank you for your valuable feedback. We have carefully incorporated all the suggested revisions into the Results section. The text has been refined to present findings objectively, without opinions or repeated analysis descriptions. Additional details on the intervention have been included for clarity. Cost estimations, intervention components, and related changes have been properly described in the Methods section. Furthermore, tables have been adjusted as per your recommendations for better readability. All changes have been made in the revised version and are highlighted in red for your reference. Please refer to the updated file.
Comments 6: Discussion
Response 6: Thank you for your insightful comments. We have incorporated all suggested revisions into the Discussion section. The role of pharmacists, clinical outcomes, and the impact of lifestyle modifications have been clearly addressed. Additionally, we have included a discussion on the strengths and limitations of the study. The section now focuses on interpreting key findings in the context of existing literature rather than repeating results. All changes have been made in the revised version and are highlighted in red for your reference. Please refer to the updated file.
|
4. Response to Comments on the Quality of English Language |
Point 1: |
Response 1: As per Respected Reviewer (The English is fine and does not require any improvement.) |
5. Additional clarifications |
With due respect, we sincerely appreciate your valuable feedback. We would like to clarify that our study was a Randomized Controlled Trial (RCT), and the primary objective was to evaluate the impact of pharmacist intervention on medication costs alongside improved clinical outcomes. The reduction in drug usage was an expected outcome of improved disease management. We politely feel that Regression Analysis was not necessary, as it was not aligned with our study’s main objective. We sincerely apologize for any misunderstanding and kindly request your consideration on this matter.
|

Reviewer 2 Report
Comments and Suggestions for Authors
All comments and revisions have been provided within the manuscript.

The English quality of the manuscript is overall fair; however, it can be improved by rephrasing some sentences.
Author Response
Comments 1: This paragraph can be moved above. (Introduction part) |
Response 1: Thank you for your valuable feedback. Changes were made as per the recommendations, and the paragraph was moved up, rephrased, and highlighted in red. |
Comments 2: Introduction and Methdology part Please provide any specific inclusion/exclusion criteria like uncontrolled hyperglycemia, hyperglycemic emergencies (DKA, HHS), diabetes type 1. - Please determine the "blinding status of the study". Please explain these interventions by pharmacists. Please calculate and report costs by USD, rather than RM. Please define "primary" and "secondary" outcomes of the study. Please define these diabetic complications. Response 2: Agree. Thank you for your valuable suggestions. We have incorporated all the requested changes into the Introduction and Methodology sections. Specific inclusion/exclusion criteria such as uncontrolled hyperglycemia, hyperglycemic emergencies (DKA, HHS), and diabetes type 1 have been clearly defined. The blinding status of the study has also been clarified. We have elaborated on the pharmacist interventions, provided costs in USD instead of RM, and defined both primary and secondary outcomes, along with the details of diabetic complications. All changes have been made in the revised version and are highlighted in red for your reference. Please refer to the updated file. Comments 3: Result Few changes suggested in Table Specifically in Table 3.2, 3.4, and 3.5. Response 3: Thank you for your valuable comments and insightful suggestions. We greatly appreciate your time and effort in reviewing our manuscript. As per your recommendations, we have made changes. All the requested changes have been made and are highlighted in red for your reference. Please refer to the updated file for the revised version. Comments 4: Discussion
Response 4: Thank you for your detailed and insightful comments. We have carefully incorporated all the suggested revisions into the Discussion section to enhance clarity and accuracy. All changes have been made in the revised version and are highlighted in red for your reference. Please refer to the updated file. |

Reviewer 3 Report
Comments and Suggestions for Authors
Thank you for the invitation email. I read the manuscript with interest. The authors evaluated pharmacists' impact on clinical management and cost reduction in patients with diabetes. I have a few comments.
- The manuscript reference format should be revised according to the journal style.
- In the method section, pharmacist intervention should be briefly discussed. In the trial registration, the author presented interventions. The authors are recommended to bring interventions in the manuscript or as a supplement.
- In the method section. Standard diabetes treatment according to which guidelines? If so, please mention the guidelines.
- The age in the intervention group was statistically higher than the control group. Please delete the following sentence. "A negligible difference was noted in the age and duration of the disease in the present investigation."
- In the results section 3.2. please bring a table to show the difference between the control and intervention groups. In addition, the authors mainly focused on medication cost and clinical data that may show pharmacist impact on diabetes management not discussed enough.
- It is better to revise the results section for better understanding. For instance, authors are recommended to use graphs to show the results and differences in a better format.
- The discussion section mainly discussed pharmacists' impact on cost reduction and did not discuss the clinical impact of pharmacists, which may be boring for the reader and should be revised.
Author Response
Comments 1: The manuscript reference format should be revised according to the journal style. |
Additional clarifications
Thank you for your thoughtful recommendation. We understand the importance of using graphs to present results in a more accessible format. However, as the main objective of our study was to evaluate the impact of pharmacist intervention on medication costs alongside improved clinical outcomes, and given that the focus was on clinical and cost measures, we feel that the use of graphs may not be necessary for our analysis. We kindly request your understanding and hope that the current presentation of the results is sufficient for clarity. Please refer to the revised version, where we have highlighted all changes in red. |

Round 2
Reviewer 1 Report
Comments and Suggestions for Authors
The authors improved their earlier manuscript, answering in detail all comments made and carefully changing the manuscript accordingly. All my prior points are met and answered to my satisfaction.
Author Response
Thank You very Much
No Further comments from this reviewer
Reviewer 2 Report
Comments and Suggestions for Authors
All my provided comments/revisions have been applied and addressed by the authors and the manuscript can be considered for publication in the current format.
Author Response

(The authors gave the same response as above.)

Reviewer 3 Report
Comments and Suggestions for Authors
Dear editors,
Thank you for the invitation email. The authors have responded to my comments.
Regards,
Author Response

(The authors gave the same response as above.)
